# The Effect of the COVID-19 Pandemic on Outpatient Antibiotic Prescription Rates in Children and Adolescents—A Claims-Based Study in Germany

**DOI:** 10.3390/antibiotics11101433

**Published:** 2022-10-18

**Authors:** Manas K. Akmatov, Claudia Kohring, Lotte Dammertz, Joachim Heuer, Maike Below, Jörg Bätzing, Jakob Holstiege

**Affiliations:** 1Department of Epidemiology and Health Care Atlas, Central Research Institute of Ambulatory Health Care, 10587 Berlin, Germany; 2Department of Prescription Data, Central Research Institute of Ambulatory Health Care, 10587 Berlin, Germany

**Keywords:** adolescents, antibiotics, children, COVID-19, claims data, Germany, prescription rates, SARS-CoV-2

## Abstract

The aim of the study was to examine whether the COVID-19 pandemic had any effect on antibiotic prescription rates in children in Germany. Using the nationwide outpatient prescription data from the Statutory Health Insurance from 2010 to 2021, changes in the monthly prescriptions of systemic antibiotics dispensed to children aged 0–14 years were examined (*n* = 9,688,483 in 2021). Interrupted time series analysis was used to assess the effect of mitigation measures against SARS-COV-2, introduced in March and November 2020, on antibiotic prescription rates. In the pre-pandemic period, the antibiotic prescription rates displayed a linear decrease from 2010 to 2019 (mean annual decrease, –6%). In 2020, an immediate effect of mitigation measures on prescription rates was observed; in particular, the rate decreased steeply in April (RR 0.24, 95% CI: 0.14–0.41) and November 2020 (0.44, 0.27–0.73). The decrease was observed in all ages and for all antibiotic subgroups. However, this effect was temporary. Regionally, prescription rates were highly correlated between 2019 and 2020/2021. Substantial reductions in antibiotic prescription rates following the mitigation measures may indicate limited access to medical care, changes in care-seeking behavior and/or a decrease of respiratory infections. Despite an all-time low of antibiotic use, regional variations remained high and strongly correlated with pre-pandemic levels.

## 1. Introduction

An increasing number of studies has shown that the COVID-19 pandemic and associated public health measures affected the epidemiology of other infectious diseases. As compared to the pre-pandemic period, the incidence of both viral and bacterial infections decreased. For example, the Robert Koch Institute in Germany reported the decrease in the number of notifiable cases for all infectious diseases, except tick-borne encephalitis [1,2]. In addition, several countries, including Australia [3], England [4], Scotland [5], Canada [6] and the USA [7], observed the reduction in antibiotic prescriptions in the early phase of the pandemic. A similar trend has been observed in Europe; surveillance data showed a reduction of antibiotic use in the general population in outpatient and hospital prescriptions combined between 2019 and 2020 of 17.6% [8]. Prior to the COVID-19 pandemic, the antibiotic prescription rate in children and adolescents had been decreasing constantly since 2010 in Germany [9,10]. Namely, the antibiotic prescription rate decreased significantly from 746 prescriptions per 1000 children in 2010 to 428 per 1000 in 2018. In the current study, we examined whether there was a similar association of the COVID-19 pandemic and associated mitigation measures against SARS-CoV-2 with the prescription of antibiotics in Germany compared to the above-mentioned countries. The first mitigation measures against SARS-CoV-2 were introduced in March 2020 and lasted two months (so-called lockdown). They comprised the closure of day-care centers, schools, leisure amenities and extensive contact restrictions (up to two persons). On 2 November 2020, a second set of mitigation measures (so-called lockdown light) was implemented and included contact reduction (up to 10 persons from the same household etc.). However, as the COVID-19 case numbers continued to rise, on 15 December 2020, the mitigation measures were extended until May 2021.

## 2. Results

In the pre-pandemic period between 2010 and 2019, the prescription rates displayed an expected seasonal pattern, with the highest prescription rates in December, January and February and the lowest rates in July. Overall, the secular trend was decreasing from 749 per 1000 children in 2010 to 401 per 1000 children in 2019, with a mean annual decrease of 6% (Figure 1a and Table 1). In 2020 and 2021, we observed a more pronounced decline (−43% and −53% compared to 2019). In particular, the prescription rate started decreasing steeply in April 2020 (9 prescriptions per 1000 children compared to 35 prescriptions in March 2020) (Figure 1a). We observed an immediate significant effect of both the first and second lockdowns on antibiotic prescription rates (Table 1). The decrease following the first lockdown was stronger (RR, 0.24; 95% CI: 0.14–0.41) than after the second lockdown (RR, 0.44, 95% CI: 0.27–0.73). The decrease in prescription rates during the COVID-19 pandemic was observed in all ages (Figure 1b,c) and for all antibiotic subgroups (Figure 2 and Appendix A). However, the effect of both lockdowns was temporary; in the months after the end of the mitigation measures, prescription rates increased (Figure 1a and Table 1). This increase in 2021, however, was age-dependent and more pronounced in children aged five years and under, and less pronounced in children and adolescents aged 6–18 years (Figure 1b,c).

In 2019, the year before the COVID-19 pandemic, the antibiotic prescription rates varied regionally by a factor of 1.9 from 294 prescriptions in Brandenburg (East Germany) to 566 prescriptions per 1000 children in Saarland (West Germany). Antibiotic prescriptions in 2019 correlated strongly with those from 2020 (Spearman’s rho = 0.95, *p* < 0.0001, Figure 3) and 2021 (rho = 0.84, *p* < 0.0001).

## 3. Discussion

The current study provides results from a near-real-life monitoring of outpatient antibiotic use among children and adolescents with the Statutory Health Insurance in Germany, who make up about 83% of the general population in this age segment. We observed an immediate effect of the mitigation measures against SARS-CoV-2 on total antibiotic prescription rates in all age groups, all German regions and for all antibiotic subgroups. In particular, the effect was more prominent following the first lockdown, which, among other measures, included the closure of kindergartens and schools, cancellation of leisure activities and extensive contact restrictions. Overall, the pediatric antibiotic prescription rate decreased by 43% in 2020 and by 53% in 2021 compared to its pre-pandemic level in 2019. These findings are not unexpected and have been observed in studies from other industrialized, including European, countries for the year 2020 [4,5,11]. A US study showed a decline in antibiotic dispensing in children aged 0–19 years of −27% in 2020 compared to 2019 [7]. A similar finding was observed in Canada; the total prescription rate decreased from 50 to 37 prescriptions per 1000 persons in all age groups, corresponding to a 27% reduction [6]. The strongest decline of about 70% was reported for children aged 0–18 years [6]. In contrast to these studies, we examined the simultaneous effect of risk-mitigation measures during the two pandemic years, 2020 and 2021, in Germany. This is important as both pandemic years displayed a shift from overall seasonal patterns of antibiotic use. In addition, seasonal patterns showed strong differences between both pandemic years.

The substantial reductions in prescription rates may be explained by limited access to healthcare facilities and/or changes in care-seeking behavior. However, large-scale mitigation measures against SARS-CoV-2 may also have resulted in the reduction of other infections, in particular respiratory tract infections of both viral and bacterial origin [12], up to the complete non-appearance of a typical seasonal outbreak of acute respiratory infections, such as respiratory syncytial virus in infants and toddlers or seasonal influenza in the winter season 2020/2021 [13]. Moreover, national notification data from the Robert Koch Institute observed the decrease in all other notifiable infectious diseases during the SARS-CoV-2-pandemic, with the exception of tick-borne encephalitis [1,2]. For example, about 163,000 infectious diseases (excluding SARS-CoV-2) were notified between March and July 2020. The numbers of the same notified infectious diseases were much higher before the pandemic (2016: 220,000; 2017: 196,000; 2018: 345,000; 2019: 248,000), corresponding to a relative reduction of about −35%. The highest decrease in notified infections was observed in children and adolescents (age group ‘0–4 years’, −57% and ‘5–14 years’, −45%) and the lowest in adults (’35–59 years’, −26%). In line with the above-mentioned studies, the strongest fall was observed for respiratory tract infections, such as measles (−86%), pertussis (−64%) and *Haemophilus influenzae* (−61%). Gastrointestinal infections due to rotavirus infection (−83%) and shigellosis (−83%) also showed strong decreases.

The re-increase of antibiotic prescriptions in 2021 compared with 2019 showed age-dependency and was less pronounced in children and adolescents older than five years compared with younger children. This might be due to age-dependent variation in health-seeking behavior on the one hand. On the other hand, it cannot be excluded that the second lockdown in 2021 was implemented stricter in schools than in kindergartens. At this point in time, it is not yet possible to judge whether the stronger decline in older children and adolescents will be sustainable in the coming years.

Of note, despite an all-time low of antibiotic use, regional variations in prescription rates remained high and were strongly correlated with pre-pandemic levels. Large-scale mitigation measures against SARS-CoV-2, especially in 2020, were, for the most part, uniformly implemented in Germany and their effect on the circulation of respiratory pathogens is unlikely to have differed between German regions. Hence, the strong correlation of regional pre- and peri-pandemic prescription rates further supports the hypothesis that pre-existing antibiotic prescription paradigms are an important explanatory factor for intra-country regional variations. Future research should assess regional differences in attitudes and levels of knowledge in the community and among healthcare providers and may inform regionally tailored interventions to further promote judicious antibiotic prescribing.

Several limitations of the study are worthy of mention. Firstly, we used secondary claims data to examine the association between risk-mitigation measures and antibiotic prescribing. This is an ecological study, which cannot establish causality. Secondly, our dataset did not contain information for antibiotics dispensed (i) by outpatient dentists and (ii) inpatient prescriptions. Finally, data for children insured privately in Germany are not part of our dataset. The latter group comprise about 13% of the general population and may differ in terms of health and socio-demographic status.

## 4. Materials and Methods

### 4.1. Data and Study Population

We used nationwide outpatient prescription data from the Statutory Health Insurance (SHI) in Germany from January 2010 to December 2021 which were collected in accordance with Article 300(2) of the Social Code Book V. The data contain all prescribed and dispensed medications (excluding dental prescriptions), the date of prescription, the pharmacy dispensation date, the amount of the prescribed substance, the anatomical therapeutic chemical (ATC) code, the defined daily doses (DDD), packaging size, strength and formulation, as well as the generic and trade names. In addition, the data include information on outpatient’s age in years and region of residence. The latter is represented by the regional Associations of Statutory Health Insurance Physicians (ASHIPs). In brief, there are 17 ASHIPs in Germany, 15 of them in each German federal state and two ASHIPs presenting in the federal state North-Rhine Westphalia. The study population comprised all children aged between 0 and 14 years in the respective years (*n* = 9,688,483 at 1 July 2021), covering approx. 83% of the total German population in this age group (Appendix A).

### 4.2. Antibiotics of Interest

We examined the prescription of the following systemic antibiotics: (i) penicillins with extended spectrum (J01CA), (ii) narrow-spectrum penicillins (J01CE, J01CF), (iii) Penicillins with beta-lactamase inhibitors (J01CR), (iv) first-generation cephalosporins (J01DB), (v) second-generation cephalosporins (J01DC), (vi) third-generation cephalosporins (J01DD), (vii) sulphonamides/trimethoprim (J01EB, J01EE, and J01EA) and (viii) macrolides (J01FA). The remaining antibiotics were categorized into the group “other antibiotics”.

### 4.3. Statistical Analysis

We calculated annual and monthly antibiotic prescription rates per 1000 children over the period 2010 to 2021, overall, as well as by age group (0–1, 2–5, 6–9 and 10–14 years), region of residence (i.e., ASHIP) and antibiotic subgroup. The total annual number of persons with SHI per age group on July 1st of a given year, derived from national statistics provided by the German Ministry of Health, was used as a denominator [14]. To examine the effect of the mitigation measures on the antibiotic prescription rate, we conducted an interrupted time series analysis using a generalized linear model with a Poisson distribution. The dependent variable was a monthly prescription rate. Since antibiotic prescribing follows a seasonal pattern, we applied seasonal decomposition to remove seasonal variation. The independent variables were: (i) time elapsed since the start of the study in months, (ii) two binary variables indicating the start of mitigation measures (i.e., first and second lockdown) and (iii) two variables for time elapsed since the introduction of each mitigation measure in months. The Spearman’s rank correlation coefficient (rho) was used to examine the relationship of the prescription rates in regional ASHIPS from the years 2019 and 2020/2021. Analyses were performed with the R Foundation for Statistical Computing, version 3.3.2 (Vienna, Austria) [15].

## Figures and Tables

**Figure 1 antibiotics-11-01433-f001:**
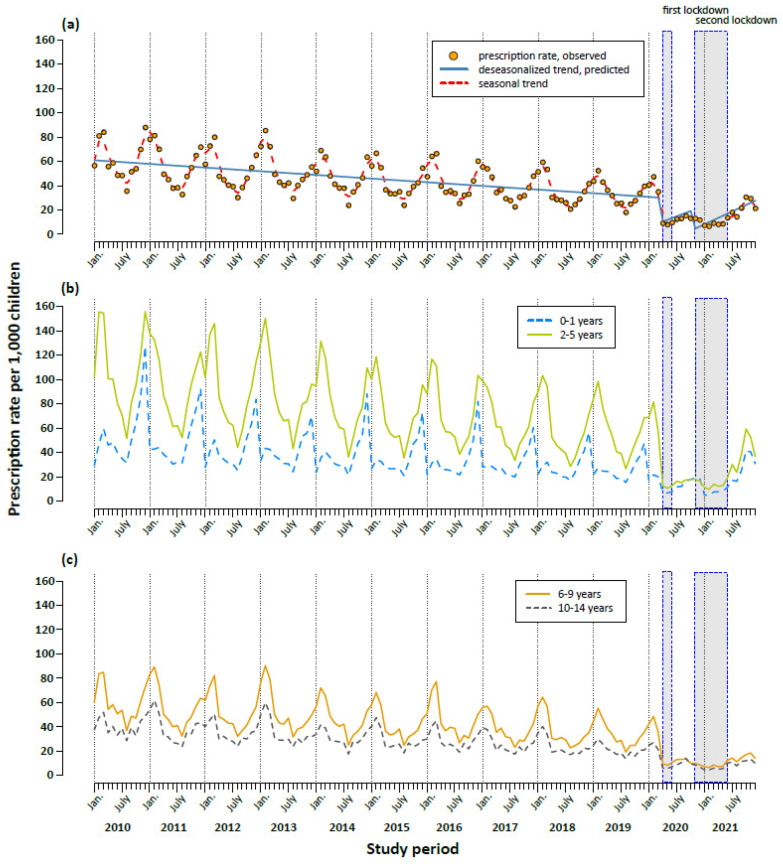
Trends in monthly antibiotic prescription rates per 1000 children aged between 0 and 14 years (**a**) and by age group (**b**,**c**) in the period 2010 to 2021. The first lockdown was introduced on 23 March 2020 and comprised extensive contact restrictions. The second, so-called lockdown light, started on 2 November 2020. (**a**) Solid line: predicted deseasonalized trend, dashed line: seasonal trend estimated with cubic spline.

**Figure 2 antibiotics-11-01433-f002:**
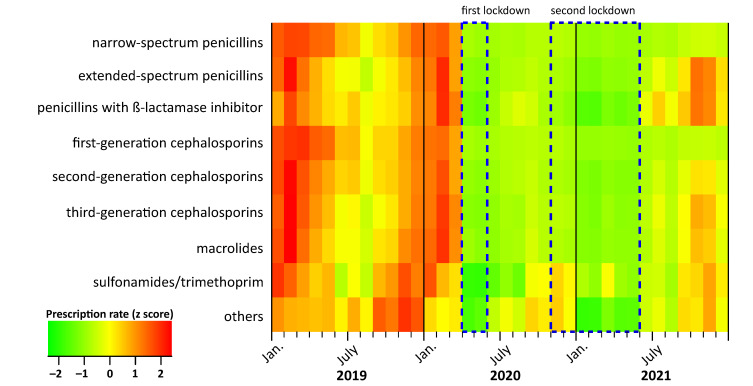
Trends in monthly antibiotic prescription rates per 1000 children aged between 0 and 14 years by antibiotic subgroup in the period 2019 to 2021. Original data can be found in Appendix A. Values for prescription rates (x) were converted into z-scores using the formula: $x\left(z-\mathrm{score}\right)=\frac{x-\mathrm{mean}\left(x\right)}{\mathrm{SD}\;\left(x\right)}$ to obtain mean = 0 and standard deviation (SD) = 1.

**Figure 3 antibiotics-11-01433-f003:**
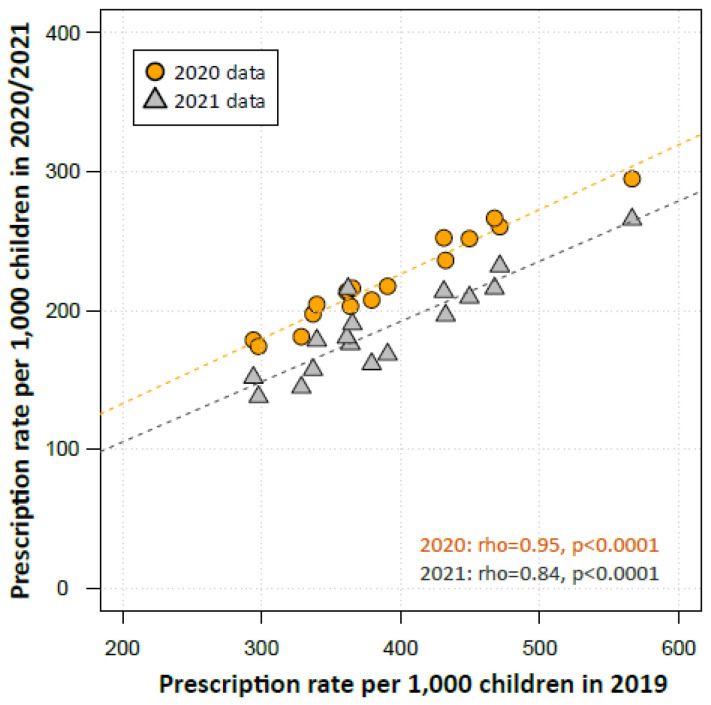
Scatter plot depicting the antibiotic prescription rates in regions in 2019 over its rates in 2020 and 2021. Regions are represented by the regional Associations of Statutory Health Insurance Physicians (ASHIP, *n* = 17). The Spearman’s rank correlation coefficient (rho) was used to examine the relationship of the prescription rates in regional ASHIPS from the years 2019 and 2020/2021.

**Table 1 antibiotics-11-01433-t001:** Effect of mitigation measures against SARS-Cov-2 on monthly antibiotic prescription rates in children in Germany—results of a generalized linear model with a Poisson distribution, 2010 to 2021.

Variables	Coefficient	Adjusted RR *	95% CI	*p* Value
Time since the study start (months)	**−0.005**	**0.995**	**0.994–0.995**	**<0.0001**
First lockdown				
Yes	**−1.422**	**0.24**	**0.14–0.41**	**<0.0001**
No		ref.		
Post-first-lockdown period (months)	0.097	1.10	0.98–1.23	0.087
Second lockdown				
Yes	**−0.812**	**0.44**	**0.27–0.73**	**0.002**
No		ref.		
Post-second-lockdown period (months)	0.009	1.01	0.90–1.13	0.873

* Adjusted for all variables in the table. Statistically significant findings are in bold. RR, relative risk; CI, confidence intervals.

## Data Availability

The datasets analyzed during the current study are not publicly available due to data protection regulations by the Social Code Book (SGB V).

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
