# Peer review of "The Effect of the COVID-19 Pandemic on Outpatient Antibiotic Prescription Rates in Children and Adolescents—A Claims-Based Study in Germany"

_antibiotics, 2022, doi:10.3390/antibiotics11101433_

Round 1
Reviewer 1 Report
Dear authors,
I have some specific comments:
1. Abstract: scientific publications usually do not use personal forms like "we". Please rephrase this part of the manuscript.
2. The design of the manuscript is difficult to follow. I recommend that you restructure it: introduction, materials and methods, results, discussion, and conclusions.
3. Figures 1,2 and 3 are illegible. Please improve their resolution.
4. Study inclusion/exclusion criteria?
5. What does your study provide in addition to other studies on this topic?
Reviewer 2 Report
The manuscript by Akmatov et al explains the effect of COVI-19 pandemic mitigation measures (specifically lockdowns) on use of antibiotics prescription rate in different age groups. The study is interesting and is continuation of work which was done by same authors during 2010 to 2018. The manuscript needs some clarifications which are mentioned as follows:
1. The authors have done data analysis from 2010 until 2021. As current year 2022 is almost in its end, whether authors can add data of prescription rates from year 2022. It will be a nice addition to look at whether antibiotic prescription rate continue to increase or decrease compared to pre-pandemic level.
2. Graph plotted in figure 1 are difficult to read because of poor resolution.
3. It is very vague to state winter month. Author should mention what months they are considering as winter months. As peaks of prescription rates in fig 1 a, either fall before, on or after the intermittent lines used for defining years, mentioning the months will add additional insights to the graph and data.
4. Y axis of Figure 1 a, b and c are not on same scale. e.g in fig 1 b the range is up to 160. It should be consistent to compare the data between fig 1 b and c.
5. The reference mentioned in supplementary table (that is ref 11) is not correct.
6. Figure 2 is not clear, how the heat map is plotted? The prescription rate is shown from -2 to 2 what does it stand for? and the unit is not clear? If the heat maps are relative function, then how it is normalized?
Round 2
Reviewer 1 Report
The authors improved the manuscript. Therefore, I agree with the publication .